# The Public Health Value of Coding Surgery in South Africa Using the International Classification of Health Interventions

**DOI:** 10.3390/ijerph20043445

**Published:** 2023-02-15

**Authors:** Sithara Satiyadev, Richard Madden, Judith Bruce

**Affiliations:** 1School of Clinical Medicine, Faculty of Health Sciences, University of the Witwatersrand, Johannesburg 2193, South Africa; 2School of Health Sciences, Faculty of Medicine and Health, University of Sydney, Sydney, NSW 2006, Australia; 3School of Therapeutic Sciences, Faculty of Health Sciences, University of the Witwatersrand, Johannesburg 2050, South Africa

**Keywords:** clinical coding, health classification, international classification of health intervention, ICHI, surgery, surgical interventions, public health

## Abstract

Background: The lack of a national intervention coding system and the current intervention coding in South Africa through the use of multiple billing and coding systems does not support quality intervention data collection and patient outcomes reporting for general surgery. This presents difficulties in the understanding of the different types of general surgical interventions performed in hospitals, the associated resources, risks, complications, outcomes reporting, public healthcare delivery, and determination of barriers to care. This study illustrates the use of accurate data on health interventions in South Africa’s acute care general surgery coding to assist in improved resource allocation, using the new WHO International Classification of Health Interventions (ICHI). ICHI has over 8000 codes and consists of three axes: Target (the entity on which the Action is carried out), Action (a deed performed by an actor to a target), and Means (the processes and methods by which the Action is carried out). A key benefit of ICHI is that it can be used collectively with the International Classification of Diseases (ICD) and the International Classification of Functioning, Disability and Health (ICF). Objectives: to evaluate the suitability of ICHI for general surgery interventions by coding intervention descriptions to ICHI codes; to identify gaps in the ICHI system; and to provide a rationale for ICHI to be a nationally regulated system. Methods: This study adopted a retrospective, descriptive design; in total, 3000 in-patient intervention data files, captured in an electronic database from April 2013 to August 2019 at three academic hospitals in Johannesburg, were extracted randomly, and coded using ICHI. Quantitative data analysis techniques were utilized to assess the overall degree of match between ICHI codes and the intervention descriptions. Results: Of the 3000 patient case entries that were coded, there was an agreement of 67.6% of the coded data amongst the three coders, leaving a variability of 32.4%. The variability was largely due to the coders’ experience and the quality of healthcare documentation. Conclusions: ICHI has the ability to cater for the broad range of general surgery interventions, thus indicating that ICHI is suitable for general surgery coding.

## 1. Introduction

Health classification systems are widely used nationally and internationally to manage healthcare delivery systems by collecting quality health data that can be communicated in a consistent, predictable, and reproducible manner. Many countries have developed national intervention coding systems that are utilized for intervention data collection and reimbursement of healthcare provider claims, such as the Canadian Classification of Health Interventions (CCI). South Africa lacks a national dataset on hospital in-patient activity, which assists with quality and safety management and resource allocation in many other countries News 24, 2022) [1].; see, for example, the use of hospital data in Australia by means of the Australian Health Intervention Classification (Madden et al., 2022) [2]. To a certain extent, the implementation of the ICD-10 has addressed the healthcare sector’s understanding of disease prevalence in South Africa and the reasons for interventions. However, the lack of a national intervention coding system and the use of billing systems, such as the National Health Reference Price List (NHRPL) as regulated by the National Health Act number 61 of 2003 regulation 1 (v), for data collection presents a challenge in the collection of hospital intervention data. Despite the evaluation of several international intervention coding and classification systems by healthcare stakeholders in South Africa (public and private), the country is yet to adopt and implement a national intervention classification system. The research aims to establish if the International Classification for Health Interventions (ICHI) [3] is suitable for coding of interventions rendered to surgical in-patients.

## 2. Methods

### 2.1. Study Design

This study adopted a retrospective, descriptive design since general surgery in-patient intervention data captured in a database were logically and rationally studied and coded using ICHI. The NHRPL codes were excluded due to a high error rate in the assignment of these codes. A quantitative paradigm was utilized to evaluate the suitability of ICHI for general surgery coding.

### 2.2. Population and Sample

The target population included datasets from surgical patient interventions at the Chris Hani Baragwanath Academic Hospital (CHBAH), the Charlotte Maxeke Johannesburg Academic Hospital (CMJAH), and the Helen Joseph Hospital (HJ). Data consisted of surgical interventions that had been performed on adult patients at all three hospitals, as well as paediatric patients from the CMJAH. The total population comprised of 30,000 discharge summaries from patient records that had been captured electronically as a pilot project by the University of Witwatersrand (Wits) to compile an electronic record.

### 2.3. Sample Size

Overall, 10% of 30,000 healthcare records were randomly selected by the database administrator for inclusion in the study (n = 3000). No power analysis was performed to determine the required sample size because no previous studies of this nature existed to establish a suitable benchmark effect size (and, therefore, sample size).

### 2.4. Inclusion Criteria

The inclusion criteria for this study were general surgery interventions performed in two academic hospitals on adults only, and on adults and children (223 records) in one academic hospital, as this hospital collected paediatric data. The sample set was limited to 10% of the population.

### 2.5. Data Collection

Data were collected, retrospectively, from patient discharge summaries into an electronic database between April 2013 and August 2019 by administrative personnel employed by Wits. The data collection was a joint project between Wits and the Gauteng Department of Health.

### 2.6. Data Management

Once collected, the datasets were exported to Microsoft Excel. Each intervention description was translated by three clinical coders (professional coder who has clinical experience; and an intermediate coder and a junior coder who are trained in coding but lack clinical experience) into corresponding ICHI codes by use of an online ICHI browser. ICHI coding rules and conventions were adhered to according to the 2020 ICHI Beta-3 Reference Guide which all three coders had access to. All three coders utilized the same datasets in the translation process. Where the intervention description was incomplete, omitted, or vague, the radiology findings and ICD-10 code/diagnostic information were referenced to assign the most appropriate ICHI code(s). A relationship key (RK) that describes the manner in which the intervention description was translated into ICHI code(s), was assigned to each ICHI code, as illustrated in Table 1.

Accuracy of the data was established by doing random spot checks on the coded data. In preparation for analysis, the data were assessed to ascertain the match between ICHI code and the intervention description, by assigning a numeric value from 1 to 7 to the key in the table above, excluding RI and II. A value of 1 indicates an exact match between the ICHI code and the intervention description, while a value of 7 indicates no match; incremental integer values between 2 and 6 represent an ordinal scale of varying degrees of match.

### 2.7. Statistical Analysis

Quantitative data analysis techniques were utilized to assess the overall degree of match between the ICHI codes and the intervention descriptions. IBM Statistical Package for the Social Sciences (SPSS) version 24 was used for the analysis of the data. Various tests were performed to present the results, some of which included the McNemar test of symmetry, which was utilized to calculate the mismatches between the 3 coders. The Cramer’s V test revealed ICHI code variations listed per case by the coders, and Cronbach’s alpha was used to determine whether there was internal consistency among the interval and dichotomous variables.

### 2.8. Ethical Approval

Permission was obtained from the Human Research Ethics Committee of Wits to undertake this study. As this was a retrospective study, informed patient consent was not possible; however, consent was obtained from the management, Heads of Departments of Surgery at the participating hospitals, and the database administrators. All records were de-identified.

## 3. Results

The complete dataset comprised 21 variable types, which were either qualitative (text strings), nominal, ordinal, or interval data variables. Each variable comprised 3000 associated data point entries, which were either single observations or blank (missing) cells relating to the clinical data of 3000 patient cases. The 10 nominal categorical variables included the patient’s age and gender, ICD-10 codes, NHRPL codes, ICHI code(s), and relationship keys represented in Table 1.

Across the 3000 patient case entries, 888 different variants of ICD-10 codes existed. The frequently coded chapters were the digestive system, injuries, and the musculoskeletal system. Other frequently observed chapters were neoplasms, and the circulatory and musculoskeletal systems. Some chapters appeared infrequently, such as diseases of the ear and eye. Whilst the NHRPL codes were not utilized in the translation of ICHI codes, they were utilized as supporting data during the data analysis works, as qualitative data appeared in the database that were unsuitable for the quantitative analysis. Of the 443 different NHRPL codes that existed across case entries, the intervention description ‘laparotomy-code 1809’ recurred in one-quarter of the mismatched cases; ‘upper gastrointestinal endoscopy-1587’, ‘endoscopic retrograde cholangiopancreatography-1778’, ‘oesophagoscopy with stricture-1579’, and ‘inguinal or femoral hernia: adult-1819’ also followed a similar pattern.

The professional coder provided the ICHI codes for the largest proportion of the 3000 patient cases (n = 2857, 95.2%), while the junior coder provided the least ICHI codes (n = 2457, 81.9%). The intermediate coder provided the ICHI codes in slightly fewer instances than the professional coder. The professional coder presented 467 different ICHI codes and translated up to four ICHI codes per patient case entry. The intermediate and junior coders presented 445 and 416 different ICHI codes, respectively, and translated up to five codes per patient case entry. The number of different ICHI code variations of 15.2% listed per case was determined by the coder who was generating the ICHI codes. Despite the variations, all three coders consistently assigned the same ICHI codes to the three topmost interventions: gastroscopy (ICHI Code: KBF.AE.AD), appendicectomy (ICHI Code: KBO.JK. AA), and endoscopic retrograde cholangiopancreatography (ICHI Code: KCO.BA.BB) represented in Figure 1.

The most common relationship key among all three coders was ‘E’, which indicated that an ICHI code was generated via a direct translation of the intervention descriptor. The professional coder stated that n = 2282 (76.1%) of the ICHI codes had been generated via RK- ‘E’, while the intermediate and junior coders noted n = 2111 (70.4%) and n = 2439 (81.4%) to be generated via RK-‘E’, respectively. The professional coder stated that 10.0% of the ICHI codes were a narrower concept, while the intermediate and junior coders stated that 1.8% and 2.9% were generated through RK- ‘N’. The professional and intermediate coder stated that 0.47% and 0.57% were generated through RK- ‘B’, whilst the junior coder stated that 10% was generated by RK- ‘B’.

There was an exact match for 67.63% of the ICHI codes between the three coders, indicating that there was 32.4% variability (skewness = −0.8). There was a 29.07% match between 2 of the 3 coders and a no match of 3.03%. The two coders with the ICHI codes that most-closely matched were the professional and intermediate coders, who displayed an 85.9% match (skewness = −2.1) between their ICHI code listings. A correlation analysis confirmed that there was a statistically significant correlation between the coders’ experience (in years), and the overall number of ICHI codes. The coders left 3.7% of entries blank where no ICHI codes were assigned. Cronbach’s alpha, when comparing the ICHI code quantities of the three clinical coders, was 0.9, which is considered to be suitable in most research applications.

## 4. Discussion

The results of this study describe the distribution of ICD-10 chapters in relation to general surgery interventions, and the accuracy of ICHI coding utilizing intervention data captured in an electronic database. The most frequently listed ICD-10 codes were from the digestive system, injury, and poisoning chapters; other frequently observed chapters were neoplasms, musculoskeletal, and circulatory system chapters. These chapters are consistent with the intervention descriptions in the database since the database consisted of interventions performed on general surgery patients. Some chapters appeared infrequently, such as diseases of the ear and eye, and were also noteworthy because diagnoses from these chapters are generally not appropriate for general surgery. The most frequently listed ICHI stem codes are for gastroscopy, appendicectomy, and cholangiopancreatography, which are consistent with the ICD-10 findings.

The reference to radiology findings, ICD-10, or diagnostic information allowed an ICHI code to be inferred where the intervention information was missing, unclear, or vague. For example, where ‘pigtail’ was documented as the intervention, and, if the ICD-10 code stated ‘abscess of the peritoneal cavity’ and the radiology comment indicated ‘percutaneous insertion of pigtail drain’, the ICHI code for percutaneous drainage of the peritoneal cavity was listed. This implies that this practice may need to continue in the initial phases of the ICHI implementation until healthcare documentation quality has improved nationally.

ICHI codes were assigned to 96.3% of the sample data included in the analysis with an exact match of 67.6% between the three coders, leaving a variability of 32.4%. Of the 32.4% variability, the coders left 3.7% of the patient entries blank. A 67.6% exact match and partial match of 32.4% suggests that the ICHI is suitable for general surgery coding as the variability is largely attributed to training issues, experience of the coders, and incomplete healthcare records. The coders left 3.7% of the patient case entries in the database blank where the RK ‘RI’ and “II” had insufficient information for the inference of an ICHI code. This finding illustrates the importance of complete healthcare documentation and highlighted the laborious coding process when healthcare documentation was incomplete. This finding depicts that healthcare documentation and coding accuracy are interlinked (Souza et al., 2018) [4]. Thus, it is incumbent on clinicians to document complete and concise information in discharge summaries that coders utilize for clinical coding (Mahbubani et al., 2018) [5].

The most common relationship key (RK) listed by the three coders was ‘E’ (equivalent), which indicated that the ICHI code could be generated through an equivalent coding of the intervention description for 76% of the intervention descriptions. As 3.7% of the patient case entries had no intervention information, the remaining 20.3% of the intervention descriptions were coded via the ‘Target’, ‘Action’ or ‘Means’ components, which illustrates that not all the interventions extracted from the database could be assigned via direct coding into an ICHI code, or at least not initially. This finding indicates that the ICHI will require refinement over time as revealed in literature reviews by different sources (Fortune et.al. 2018) [6] (ICHI Beta-3 Reference Guide, 2020) [7].

Less than 15% of the codes assigned by the coders had a RK of ‘N’; for example, open reduction of the knee joint with internal fixation is described by two ICHI codes, i.e., ‘MMJ.LD. AA: Open reduction of knee joint’ and ‘MMJ.DN. AA: Implantation of device into knee joint’. This will present a challenge for users that are familiar with coding one code to report both interventions. However, it must be noted that the ICHI deliberately avoids compound intervention coding and has the capacity to code several interventions and link them. The ICHI is a work in progress as it must provide a stable structure and basis for collecting data to support comparability and to serve the needs of its users (ICHI Beta-3 Reference Guide, 2020).

Some variability can be attributed to the coder’s experience. The professional coder only assigned the ICHI codes that described the surgical intervention, as the diagnostic intervention is included in surgical intervention if performed during the same operative session for the same anatomic site. Conversely, the intermediate and junior level coders assigned the ICHI code for the diagnostic intervention. For example, where the intervention description stated ‘ERCP with drainage’, the professional coder assigned the ICHI code ‘KCO.JB.AD: Endoscopic drainage of pancreas’, which includes the ERCP. The intermediate and junior coders assigned the ICHI code ‘KCO.BA.BB: Endoscopic retrograde cholangiopancreatography’, which is a diagnostic intervention code. This was noted in the rest of the mismatched case entries. Coder experience also created variability in cases where the RK was either RI or I, as reflected in Table 2.

The professional coder coded the intervention description ‘laparotomy’ to the ICHI codes that described interventions such as appendicectomy, whereas the intermediate and junior coders translated ‘laparotomy’ to an equivalent ICHI laparotomy code. Intervention coding rules state that the main intervention performed in the abdomino-pelvic region, such as appendicectomy, should be coded instead of the laparotomy, which is the means to gain access to the appendix. These findings reveal that no new ICHI codes are required to address the general surgery coding needs in South Africa. Instead, the quality of healthcare documentation and coder training requires attention.

ICHI requires refinement in some areas. For example, in proctoscopy with biopsy-on searching ‘protoscopy biopsy’, no ICHI code was found. The coder had to access the Target component, rectum, to arrive at the correct ICHI code: ‘KBW.AD.AD: Endoscopic biopsy of rectum’. Following the search via the Target component, the coder searched for biopsy of rectum and arrived at the same code. Proctosigmoidoscopy appears in the ‘Includes notes’ of the ICHI, however, proctoscopy is excluded. Similar challenges were experienced in other instances, for example ‘wedge resection-stomach’ was searched via the Target access to arrive at the ICHI code ‘KBF.JJ. AA: Partial gastrectomy’. ‘Wedge resection’ does not appear in the ‘Index Terms’ or ‘Includes notes’ of ‘partial gastrectomy’. This has implications for the search functionality of the ICHI platform.

The results highlighted the variability in coding between the coders and confirmed that, unless coders were adequately trained, they will provide variability and inconsistency in their ICHI codes. The variability can be attributed largely to the sequencing of the ICHI codes, whereby coders selected different primary ICHI codes for the same case entry, which highlights the need for training and ICHI coding standards. This is supported in literature by a study performed by Stojanovic et al. (2020) [8] to assess ICHI, which found that a period of training was necessary to understand the ICHI platform and rules. It is worth noting that the variations in assignment of the ICHI codes and relationship keys between the coders are procedural rather than a gap in the ICHI system itself.

### Study Limitations

The quality of healthcare records is a significant limitation of this study, which resulted in the coders referencing the ICD-10 codes/radiology information to make inferences in the assignment of some ICHI codes. A further limitation was the failure to assign ICHI codes to some case entries as the intervention descriptions were missing, and it was not possible to make inferences from ICD-10 codes and radiology findings. The exclusion of ICHI extension codes, which are optional, may have provided different results in respect to the assignment of relationship keys.

The important findings between the experience and training of the coders and the assignment of codes reflects that, unless a coder is adequately trained and experienced, variability is likely to occur. Utilizing professional coders in the assignment of ICHI codes may have presented a different outcome. The researcher utilized coders with different levels of experience to reflect the current coding practice in South Africa, which presents evocative results for policy considerations. Another limitation was the assignment of the relationship keys by the coders, which indicates that procedural matters require attention to avoid a greater degree of variability amongst a wider range of coders. Furthermore, the association between the dependent and independent variables may be confused by dynamics that were not investigated in this study. The study did not include health data from all hospitals in South Africa, and the inclusion of general surgery data limits the possibility of the results to be generalised.

## 5. Conclusions

The application of international health classifications is essential to provide the knowledge base to allocate limited resources (human and financial) as efficiently and effectively as possible in South Africa. Adaptive use of classifications will be essential given the limitations in data collection and coding skills in the country at present.

This research has shown the strengths and challenges of ICHI through a review of the literature and an analysis of retrospective general surgery intervention data. Detailed illustration of the various challenges and limitations in the assignment of ICHI codes and recommendations makes a valuable contribution to the adoption and implementation of the ICHI as a national intervention coding system for general surgery. This study makes a significant contribution to evidence-based knowledge as no ICHI research of this magnitude currently exists for acute care general surgery coding.

Coder training, accreditation, and experience is integral in decreasing coding errors and coding variabilities amongst coders (Dyers et. al., 2016) [9]. The results indicate that trained and more experienced coders are less likely to make coding errors as opposed to less experienced coders that are not adequately trained. National ICHI coding standards and guidelines and improved healthcare documentation will further assist in decreasing variabilities and coding errors. ICHI can replace the current intervention data collection systems in use in South Africa, but not the multiple billing systems in its current form. For hospital billing, an ICD-10/ICD-11 and ICHI Diagnosis Related Grouper can be developed. The creation of relative value units can evolve the ICHI into a billing system for medical, allied, and support health professionals. To this end, the ICHI does have the potential to replace all the multiple coding and billing systems in use in South Africa supported by appropriate national policies and regulations.

## Figures and Tables

**Figure 1 ijerph-20-03445-f001:**
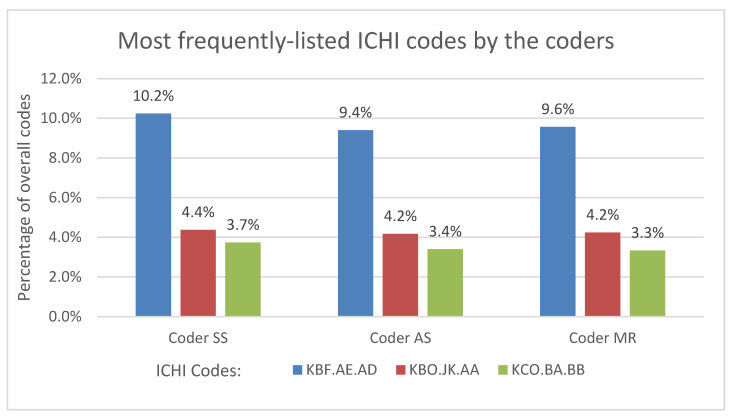
Most frequently listed ICHI codes by coders.

**Table 1 ijerph-20-03445-t001:** Relationship keys for coding of intervention description to the ICHI codes.

Key	Meaning	Explanation
E	Equivalent translation	One-to-one: one intervention description to one ICHI code.
B	Broader concept	Many-to-one: many intervention descriptions to one ICHI code.
N	Narrower concept	One-to-many: one intervention description to many ICHI codes.
TM	Target component translation	Intervention descriptor translates to the ICHI Target component.
AA	Action component translation	Intervention descriptor translates to the ICHI Action component.
AM	Means component translation	Intervention descriptor translates to the ICHI Means component.
NC	No corresponding code	No corresponding ICHI code for intervention descriptor.
NI	No intervention information	Procedure field in database was blank.
RI	Radiology information	ICHI code was assigned on radiology report as the intervention field was blank.
II	ICD-10 information/diagnostic information	ICHI code was assigned on ICD-10 code/diagnosis description as intervention information was incomplete or vague.

**Table 2 ijerph-20-03445-t002:** Variability for relationship keys RI and II.

Coders	RK: RI	RK: II
Professional coder	n = 165 (5.5%)	n = 44 (1.5%)
Intermediate coder	n = 110 (3.7%)	n = 1 (0.03%)
Junior coder	n = 228 (7.6%)	n = 39 (1.3%)

## Data Availability

Available from the University of Witswatersrand on request.

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
