# Peer review of "The Public Health Value of Coding Surgery in South Africa Using the International Classification of Health Interventions"

_ijerph, 2023, doi:10.3390/ijerph20043445_

Round 1

Reviewer 1 Report

Thank you for your submission. What is unclear to me is how this adds to the body of knowledge. ICHI has been routinely used for more than a decade for general surgery in many countries.How does your research inform those practices?

Author Response

Dear Reviewer

Kindly find attached responses from the authors and the updated manuscript has been uploaded with changes highlighted in yellow.

Kind Regards

Sithara

Reviewer 2 Report

REVIEWER’S COMMENTS

The Public Health Value of Coding Surgery in South Africa 2 Using the International Classification of Health Interventions

Comments to Author:

1.    In Background section

The manuscript deals with an essential subject concerning the coding of health care procedures to support decision support particularly that related to billing. The implementation of a standardized coding support system will not only help to improve the billing of medical procedures, for example in surgery interventions, but also to improve the collection of quality data.  The authors therefore address a major public health concern. The context in general is very edifying, except that I have trouble perceiving in the manuscript the state of play on Narcotics in South Korea. If this is not done, it will be desirable for the authors to add some in the context.

However, in the absence of a state of the art section on medical procedure coding systems for decision support, it would be desirable for the authors to make the context as exhaustive as possible by succinctly adding a state of the art of coding systems at the global level (for example in Europe, Asia, Africa, etc.) and then presenting the South African context.

2.    In method section

** Some clarifications needed

-       I assume that the abbreviation "ICHI" refers to the International Classification of Health Interventions? If so, please write in the text for the first use.

-       Is there a methodological reason for choosing only the three teaching hospitals?

-       Why did you choose to collect data from pediatric patients only from CMJAH, whereas for adults, data were collected from all three teaching hospitals?

-       Had the administrative staff employed by the University of the Witwatersrand and responsible for collecting the data received prior training on the variables to be collected and their definition? Isn't there a risk of selection bias? 

-       All records were anonymised prior to data collection by administrative staff? In the event that the anonymization of the files is done after collection, how do you guarantee confidentiality with the administrative staff who collected the data?

** Observations

-       Regarding sample size, You would have been able to keep a sample from the individuals in your population of 30,000 records that met your inclusion and exclusion criteria. A reasoned choice of 10% may compromise the power of the results, knowing that among the 90% not retained there may be individuals who meet the conditions of eligibility in your study and may lead to different results.

3.    Results

On page 4 lines 135 to 137 in results sections, you say: “There was a statistically normal distribution of age ranging from 0 to 100 years; 55.6% of the healthcare interventions were performed on males and 44.4% on females.”

It doesn't seem clear when you talk about a statistically normal age distribution from 0 to 100 years. No parameters (average, median mode are mentioned in both groups).

Even mentioning percentages by sex this is still not sufficient to support that there is a statistically normal age distribution.

This sentence will have to be reworded.  

After the above, I suggest that the manuscript be accepted with some minor revisions.

Author Response

Dear Reviewer

Attached please find comments from the authors highlighted in yellow. The updated manuscript with changes highlighted in yellow has been uploaded for your perusal.

Kind Regards

Sithara

Reviewer 3 Report

Thank you very much for the opportunity to review the manuscript. It describes work that has the potential to make a significant contribution in South Africa and areas of the developing world that struggle with similar problems regarding data collection and data use in the surgical patient population.

General comments:

The manuscript was submitted as a protocol, which is inappropriate. I would suggest submitting a revision of the manuscript as an Article or Project Report. The comments that follows are aimed at guiding such a new submission, and no word count checking was done.

In general, the manuscript can be significantly improved by providing more information on the context in which the study was carried out. E.g., the authors make little reference to the difference between the use of coding for interventions in the public and provide sectors in the heavily fragmented SA healthcare system, and to established efforts in the country to address problems with data collection and data use (e.g. only an abbreviated reference to the ‘WCDoH eCCR’ in the discussion. It also fails to explain up-front the role of clinical coders (the assumption is made that the ‘researchers’ and/or authors are clinical coders) in the larger context of healthcare data use. If presented appropriately, this work can serve as a roadmap to implementing ICHI not only in SA, but in other developing or low-and middle-income countries, and therefore have more impact.

Title

The title indicates a focus on ‘Public Health Value’ and does not reflect the study aim (‘to establish if ICHI is suitable for coding of interventions rendered to surgical in-patients’) nor the study population of what is assumed to be public sector surgical patients. The study appears to be a feasibility study for the implementation of ICHI, rather, and I suggest that the authors consider amending the title. It is good practice to ensure that the article title is similar to the study title, unless a different type of manuscript than an Article is chosen.

Abstract

The background statements in line 11 to 13 is not factually correct and seems somewhat inflammatory. Please refer to comments below on the Introduction for further explanation.

The Background section of the abstract seems long in comparison to the other sections. I would suggest cutting down on the background section and providing more detail in the Methods and Results section.

Introduction

There is no doubt that a universal standard for coding of interventions (including surgical procedures) using a system such as ICHI has significant benefit in public health – particularly in the Global Surgery field, where advocacy for surgical system strengthening is so important. Yet there is almost no reference in this section of the manuscript on the work that has been done in either the Global Surgery field or coding initiatives, locally in South Africa or internationally. Colleagues in South Africa that have been active in these areas include representatives from both the private and public sector, and some have contributed to task teams established by the SA National Department of Health. The authors’ only reference in this section is a self-reference, and the Introduction therefore provides little context to the work that was done in the study – which ultimately will significantly lessen the impact of a publication.

Methods

Line 69 – ‘purposively selected’: what was the purpose of selecting these hospitals? CHBAH – this is the first reference to the hospital in the text, so the full name should be provided with the abbreviation in brackets.

Line 71 – why was the paediatric patient population excluded at 2 of the 3 hospitals, and what is the relevance?

Line 76 – what was the decision to inspect 10% of the records based on, and how where these records selected?

Line 81 – refer to question on exclusion of paediatric patients above

Line 83 – ‘exclusively to interventional data’ Patient age and gender was included in the dataset, as stated in line 129. This statement is therefore not correct?

Line 95 – which of the 3 authors are ‘the researcher’? Please clarify the role of ‘researcher’.

Line 115 – please specify which tests were used for which types of data. More information is required to ensure reproducibility of results.

Results

Line 129 – NHRPL codes should be introduced and explained in the Introduction

Line 132-136 – these results should be explained by the Methods section on sampling and inclusion criteria, e.g., the range and individual contribution of surgical disciplines at the different hospitals that had data available.

Line 138 – Please explain elsewhere (Introduction or Methods) why ‘NHRPL codes’ were not used in the translation to ICHI, since this system is the most widely used procedural coding system for South African clinicians. A major obstacle to implementation of ICHI would be how user-friendly it would be for application in claims data. My opinion is that the authors should consider a pragmatic approach to their recommendations on the implementation of ICHI, and ‘billing’ using ICHI would certainly be an incentive for using ICHI in the relatively well-resourced SA private sector.

Line 147,148 – More information about the usual practice of the ‘junior and intermediate coders’ under Data Management in the Methods section may contribute to a better understanding/interpretation of these results. It is assumed that none of these coders have experience of clinical practice or have been responsible for creating the discharge summary records, and the role of coders can be explained in the Introduction, for the benefit of a clinical audience.

Discussion

Line 187 – this (General Surgery) is the first mention of the specific surgical discipline of the records used. It must be stated in the Methods section.

I believe the Discussion and Conclusion contains valuable information and it is the main reason that I would encourage the authors to resubmit this work for publication. These sections are however somewhat unstructured, and perhaps too long.

Author Response

Dear Reviewer

Attached please find comments from the authors highlighted in yellow. The updated manuscript has been uploaded with changes highlighted in yellow.

Kind Regards

Sithara

Round 2

Reviewer 1 Report

It is improved.

Author Response

Dear Reviewer

Please find updated comments

Thanks

Sithara

Reviewer 2 Report

Overall, the paper has been improved. It lacks of references as various works, initiative are mentioned without any proper citation. For example, National Health Reference Price List has no any reference. "Many countries
have developed national interventions coding systems that are utilized for intervention
data collection and reimbursement of healthcare provider claims."  " some references of such a work should be indicated.

The first use of ICHI should define the acronym in the introduction, although the Abstract did it.

Author Response

Dear Reviewer

Attached please find updated comments

Thanks

Sithara
